# Sulfur Sources Mediated the Growth, Productivity, and Nutrient Acquisition Ability of Pearlmillet–Mustard Cropping Systems

Sanjay Singh Rathore [1], Subhash Babu [1,*], Vinod Kumar Singh [2], Kapila Shekhawat [1], Rajiv Kumar Singh [1], Pravin Kumar Upadhyay [1], Mohammad Hashim [3], K. C. Sharma [4], Rameti Jangir [1] and Raghavendra Singh [5]

1   Division of Agronomy, ICAR-Indian Agricultural Research Institute, New Delhi 110 012, India
2   ICAR-Central Research Institute for Dryland Agriculture, Hyderabad 500 059, India
3   ICAR-Indian Agricultural Research Institute, Samastipur Centre, Samastipur 848 125, India
4   ICAR-Indian Agricultural Research Institute, Indore Centre, Indore 452 001, India
5   ICAR-Indian Institute of Pulses Research, Kanpur 208 024, India
*   Correspondence: Subhash.Babu@icar.gov.in

**Abstract:** Globally, excess soil nutrient mining is a serious challenge to sustainable agricultural production. The task is more daunting in the semi-arid region of the globe. In addition to the primary nutrient deficiency over the mining of secondary nutrients, especially sulfur is an emerging challenge in contemporary crop production systems. Hence, there is a dire need to devise an appropriate sulfur management protocol by searching for efficient and sustainable sulfur sources. Therefore, the efficacy of new sulfur molecules was evaluated on the performance and nutrient acquisition potential of the pearl millet–mustard system at the research farm of the Indian Agricultural Research Institute in New Delhi, India, for two years. The flexibility of urea–ES technology allows customized sulfur-enhanced urea formulations that suit unique crop needs, offering an all-in-one nitrogen and sulfur fertilizer solution. Hence, the study hypothesized that new sulfur molecules like sulfonated urea (SEU) will have a positive impact on crop growth, yield, and nutrient acquisition in the pearl millet–mustard system. The results revealed that the application of 50% sulfur (S) (15 kg/ha) as a base and 50% (15 kg/ha) as a topdressing (SEU 10-0-0-75) produced better growth, yield-contributing parameters, and economic productivity of the pearl millet–mustard system. However, both compositions of SEU (40-0-0-13 and 10-0-0-75) were significantly better than the recommended dose of fertilizer (RDF) and the RDF along with other S sources like single super phosphate and bentonite in enhancing crop growth and productivity. The agronomic efficiency of nitrogen (AEn) and S (Aes) of SEU (40-0-0-13 and 10-0-0-75) were 9.1 and 10.3 kg seed yield/kg N and 51.2 and 28.9 kg seed/kg, respectively. The agronomic nitrogen use efficiency of SEU (40-0-0-13) and SEU (10-0-0-75) is significantly higher than those of nitrogen, phosphorus, and potassium (NPK) alone. Thus, the findings inferred that splitting S (50% as a base and 50% as topdressing) through SEU is a practically feasible and economically robust S option for profitable and sustainable production of the pearl millet–mustard production model in the semi-arid region.

**Keywords:** nutrient use efficiency; oil seeds; sulfonated urea; soil nutrient balance

## 1. Introduction

Humanity faces enormous emerging challenges, and many of them are directly associated with the food security of the exploding population, which is expected to cross the 11-billion mark by 2050 [1]. A sustainable food system depends largely on healthy soil, as soil is the basis of a healthy food system; healthy soils produce healthy crops that in turn nourish people and animals [2]. Undeniably, soil quality is directly linked with food quality and quantity [3]. In the 20th century, innovations in farm technologies helped the rapid growth of crop productivity to feed the burgeoning population; amongst those, the

use of mineral fertilizers had an incomparable impact on crop production [4]. However, escalating nutrient deficiencies have been recognized as one of the major reasons for the deteriorating soil health and productivity of predominant cropping systems in the Indian sub-continent, with sulfur (S) deficiency being one of them [5]. Furthermore, NPK-based nutrient management approaches snub the application of other nutrients, which has led to a situation where the application of nutrients like S decides the productivity level of intensive cropping systems, as the response to NPK fertilizers is muted under many situations [4,6,7]. S is a vital nutrient for plant growth and development, as it is involved in amino acid and protein synthesis [8]. Research reports showed that ~46% of soils are deficient in S, while 30% are medium in available S—which could be considered potentially S-deficient—across different states of India [9,10]. In comparison to nitrogen (N), plants take about a tenth of the amount of S; hence, a continuous supply of S is important for healthy plant growth during plant ontogeny. Declining S and N use efficiency is an emerging crisis for sustainable crop production [11]. Major factors leading to S deficiency are the inherently low S content of the soil, coarse sandy texture, low organic matter content, and conditions that favor leaching losses of available S [12,13]. The oilseeds have higher S requirements compared to cereals and other crops; not only the seed yield but also the oil content declines due to inadequate S supply in oilseed crops [14]. Owing to the synergistic effect of N × S, adequate S supply determines the N use efficiency [11]. Adequate S supply mediates the enzyme activities involved in N metabolism (nitrate reductase and nitrite reductase) in plant systems [15], and hence S deficiency can lead to a decline in N assimilation. The S and N use efficiency in field crops rarely exceeds 12% (8–12%) and 40%, respectively. N is a highly mobile nutrient in the soil, and hence leaching N losses beyond the root zone are a big issue with chemical fertilizers [16,17]. The blending of S with urea (an N source) facilitates the slow release of N and thus reduces N losses. S is an immobile element within the plant system; therefore, an incessant S supply is obligatory from emergence to crop maturity for growth and yield. S deficiency at any crop stage results in reduced growth and productivity [18]. S fertilizer applied to a crop is not utilized completely by that crop, and the S left in the soil exhibits a residual effect on the growth and yield of subsequent crops. The nutrient supply needs to be synchronized with the crop demand to reduce nutrient losses and improve nutrient use efficiency. Hence, we hypothesized that the use of new S molecules and their innovative application could potentially overcome the problem of S deficiency in contemporary crop production. S-enhanced urea (40-0-0-13 and 10-0-0-75), along with other S fertilizers like single super phosphate (SSP), bentonite, and ammonium sulfate were evaluated in the present study to find the best S source with suitable splitting for enhanced productivity and nutrient use efficiency for the overall sustainability of the pearl millet–mustard system under semi-arid conditions.

## 2. Materials and Methods

### 2.1. Site Description

To find out the best sulfur (S) source for profitable and soil-supportive pearl millet–mustard production in semi-arid climatic conditions, the experiment was conducted from 2020–2022 at the research farm of ICAR—the Indian Agricultural Research Institute—in Pusa, New Delhi. The research field is located at latitude 28°38′23″ N, longitude 77°09′27″ E, and 228.61 m of altitude. The soil of the experimental site was sandy loam in texture, neutral to slightly saline in reaction, and low in soil organic carbon (SOC), with available N and P while being medium in S content. The weather was moderate during both the crop growing seasons. During normal weather conditions, the rainfall received was 708.7 mm, while during the experimentation period (2020–2021 and 2021–2022), rainfall was 948.9 and 1781.4 mm, respectively, which was more than during a normal monsoon year. As monsoon season prevails for three months only (July to September), up to 85% of the total annual rainfall was received during these three months (Figure 1) (July, August, and September).

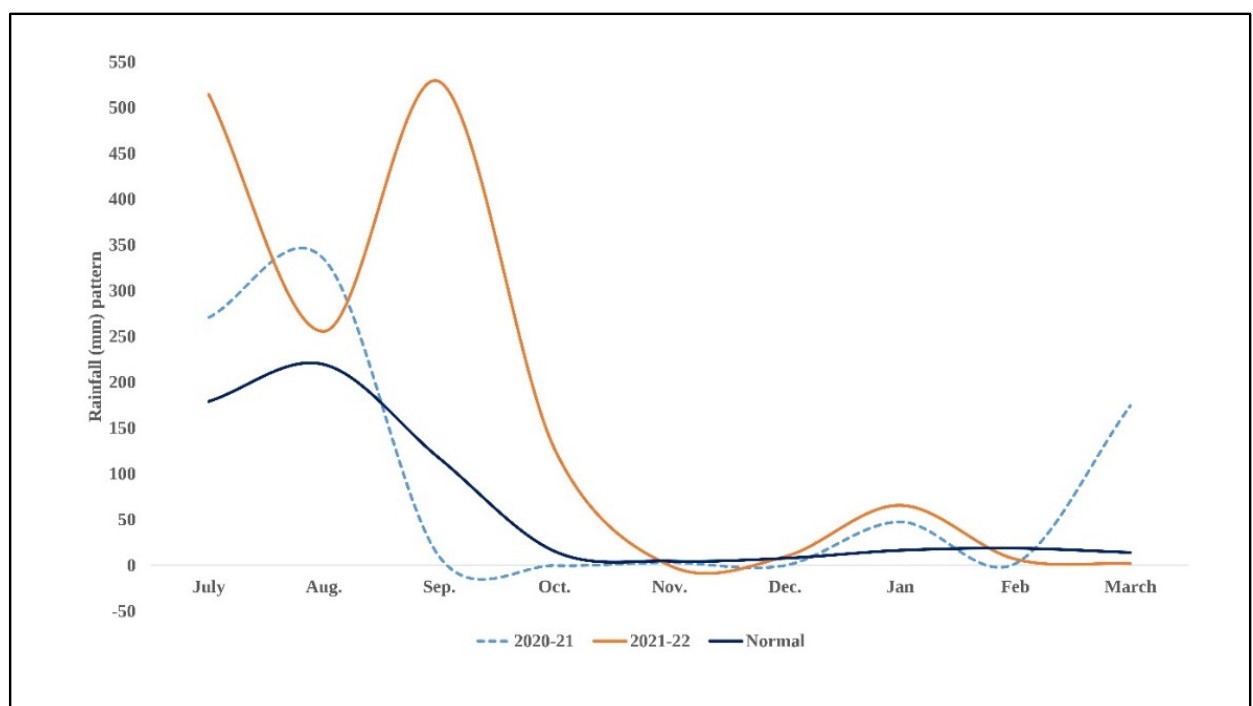

**Figure 1.** Mean monthly weather parameters recorded throughout the crop seasons of experimental years 2020–2021 and 2021–2022.

### 2.2. Experimental Detail

The experiment was carried out during the *Kharif* and *rabi* seasons of 2020–2021 and 2021–2022 in a randomized block design (RBD) comprising 12 treatments, viz., $T_1$ (no sulfur (S), recommended dose of nitrogen, phosphorus, and potassium (NPK) only), $T_2$ (recommended dose of NPK with S through 40-0-0-13 (sulfur enhanced urea: SEU) to both crops), $T_3$ (recommended dose of NPK with S through 10-0-0-75 (SEU) to both crops), $T_4$ (recommended dose of NPK with S through ammonium sulfate to both crops), $T_5$ (recommended dose of NPK with S through bentonite S to both crops), $T_6$ (recommended dose of PK and S through single super phosphate (SSP) to both crops), $T_7$ (recommended dose of PK and S through SSP to both crops), $T_8$ (recommended dose of fertilizers (RDF)+S (30 kg/ha) applied to the first crop and residual effect in the second crop based on $T_2$ source), $T_9$ (50% S (15 kg/ha) as the base and 50% (15 kg/ha) as a topdressing based on $T_2$ source), $T_{10}$ (30 kg/ha S applied to the first crop and residual effect in the second crop based on $T_3$ source), $T_{11}$ (RDF+50% S (15 kg/ha) as the base and 50% (15 kg/ha) as topdressing based on $T_3$ source), and $T_{12}$ (RDF+60 kg S/ha applied to the first crop for entire cropping season based on $T_3$), with three replications. The size of individual plots was 5 m × 4 m (20 m$^2$). The RDF for pearl millet and mustard was 120:40:40:30 kg/ha for N: $P_2O_5$: $K_2O$: S. Mustard was sown manually by the pora method.

### 2.3. Observation of Growth and Yield Attributing Parameters

The growth and yield attributing parameters of pearl millet, viz., plant height, effective tillers, ear length, ear weight, and grain weight per ear, were recorded from five randomly selected plants. Similarly, the growth and yield attributes of mustard, viz., plant height, primary and secondary branches, and total siliqua/plant, were also recorded from five randomly selected plants. Silique length and seed/silique were recorded from 20 siliques selected from five plants. Dry matter accumulation per plant was recorded from five selected plants for both crops. After air drying, the samples were placed in an oven at 65 ± 1 °C temperature. After oven drying, the average dry matter weight was recorded

and expressed in grams per plant. Treatment-wise, 1000 seeds of pearl millet and mustard were counted with seed counterweighting and expressed in grams.

### 2.4. Harvesting and Yield Measurement

At physiological maturity, the border lines were harvested first and removed from the experimental area. Thereafter, plot-wise ears of an individual plot of pearl millet were harvested first, and then the remaining crop stalks were harvested near the ground by sickle and dried. After sun drying, weights were recorded and expressed in tonnes per ha. Similarly, for the mustard crop the net plot area was harvested, and the produce was allowed to sun dry in the respective plots for three days. Thereafter, the produce was tied into bundles and weighed for recording the total quantity of produce separately for each plot. Plot-wise threshing and cleaning operations were carried out for both crops. Subsequently, the grain yields were recorded, and stover yields were computed by subtracting the grain yields from the total biological yield. Total biological, grain, and stover yields were expressed in t/ha. To measure the effect of S sources on the reproductive efficiency of pearl millet and mustard crops, the harvest index (HI) was calculated with the following expression:

$$\text{HI} = \frac{\text{Grain yield (t/ha)}}{\text{Biological yield (t/ha)}} \times 100$$

### 2.5. Oil and Fatty Acids Estimation

Treatment-wise, a 2 g seed of mustard was dried, ground into small particles, and placed in a porous cellulose thimble. Thereafter, the oil content of the seed was determined by an automatic Soxhlet extractor. The oil content in seeds was expressed as a percentage. The saturated fatty acids (SFA), monounsaturated fatty acids (MUFAs), and polyunsaturated fatty acids were estimated by standard protocols. The oil yield was computed for each treatment by using the following formula:

$$\text{Oil yield (kg/ha)} = \frac{\text{Oil content in seed (\%)} \times \text{ seed yield (kg/ha)}}{100}$$

### 2.6. Plant and Soil Analysis

Plant and soil samples were analyzed at the Crop Science Laboratory, Division of Agronomy, Indian Agricultural Research Institute, New Delhi. Plot-wise, grains and stover of pearl millet and mustard were collected from the threshed produce, dried, and ground for laboratory analysis. The N and S contents of the plant materials were analyzed by the procedure outlined by Prasad et al. in 2006 [19]. For the soil analysis, composite soil samples were drawn from the plow layer (0–15 cm depth) before commencement and at the end of the experiment. The soil samples were dried under shade, ground, and then sieved through a 2 mm sieve. Soil chemical properties like pH, electrical conductivity (EC), organic carbon, available N, and available sulfur were estimated as per the protocols outlined by Prasad et al. in 2006 [19].

### 2.7. Nutrients Acquisition and Use Efficiencies

To assess the effect of different S sources on the nutrient uptake ability of the pearl millet–mustard system, the nutrient (N and S) acquisitions by economic product (seed/grain) of both the crops were worked out by using the following formula:

$$\text{Nutrients uptake (N/S)} = \frac{\text{Nutrient (N/S) concentration (\%)}}{100} \times \text{economic yield (kg/ha)}$$

The agronomic use efficiency and partial factor productivity of nitrogen and sulfur are measures of N and S uptake regarding the quantity applied. These are imperative indicators of economic efficiency and environmental sustainability, as both show a strong relationship between the input applied (N and S) and the economic yield. The agronomic use efficiency and partial factor productivity of nitrogen and sulfur were estimated by the following formulae:

$$\text{Agronomic use efficiency} = \frac{\text{Grain yield (kg/ha) in treated plot} - \text{Grain yield (kg/ha) in control plot}}{\text{Nutrient applied (kg/ha) in treated plot}}$$

$$\text{Partial factor productivity} = \frac{\text{Grain yield (kg/ha)}}{\text{Nutrient applied (kg/ha)}}$$

### 2.8. Statistical Analysis

Randomized block design (RBD) was used to analyze the data obtained from both crops by analysis of variance (ANOVA). Before performing the ANOVA, the homogeneity of variance of all characteristics was tested by using Bartlett's tests. Comparison of means was performed with the critical difference (CD) procedure ($p \leq 0.05$). Pooled analysis of the two years was worked out as per the method described by Panse and Sukhatme [20], and Pearson correlation analysis was performed by using the software SAS (version 9.3), Cary, NC, USA.

## 3. Results

### 3.1. Growth and Yield Attributes

Sulfur application had a significant impact on the performance of the pearl millet–mustard cropping system, and the increase was significantly higher under SEU (40-0-0-13 and 10-0-0-75) than for NPK and other S sources (Table 1). Furthermore, increases in plant height, effective tillers, ear length, weight, and 1000 seed weight were observed in the pearl millet crop, along with a similar increase in growth and yield attributes in the mustard crop. The split application of SEU (10-0-0-75) was observed to be associated with maximum increases in plant height (230.5 and 178.89 cm) of pearl millet and mustard, respectively. Maximum effective tillers (2.70), ear length (28.47 cm), weight per ear (31.54 gm), and seed weight/ear (21.49 gm) of pearl millet; and maximum PB (6.26), SB (13.34), total siliqua (514.1), silique length (5.59 cm), and 1000 seed weight (4.30 gm) of mustard were recorded under $T_{11}$, but it remained on par in S management with $T_9$ and $T_3$. However, $T_6$ resulted in the statistically lowest weight/ear and grain weight/ear among the remaining treatments, whereas the application of S through SEU (10-0-0-75) ($T_3$) remained statistically at par, barring $T_{10}$, $T_8$, $T_7$, $T_5$, $T_4$, $T_1$, and $T_6$. Conversely and significantly, the lowest values of all growth and yield parameters were recorded in the plot that received no nitrogen ($T_6$).

**Table 1.** Effect of S-enhanced urea and other S sources on yield attributes and yield of pearl millet and mustard.

| Treatment | Pearl Millet | | | | | | Mustard | | | | | | |
|---|---|---|---|---|---|---|---|---|---|---|---|---|---|
| | Plant Height (cm) | Effective Tillers/ Plant | Ear Length (cm) | Weight /Ear (g) | Grain Weight/Ear (g) | 1000 Seed Weight (gm) | Plant Height (cm) | Primary Branches/ Plant | Secondary Branches/Plant | Silique Length (cm) | Total Silique/Plant | Seed/Silique | 1000 Seed Weight (gm) |
| $T_1$ | 195.9 | 1.92 | 22.07 | 22.10 | 16.27 | 4.85 | 163.67 | 4.60 | 8.60 | 3.71 | 322.7 | 11.43 | 3.33 |
| $T_2$ | 218.9 | 2.43 | 26.76 | 28.51 | 18.37 | 5.30 | 168.70 | 5.54 | 12.11 | 5.29 | 428.7 | 13.03 | 3.70 |
| $T_3$ | 218.7 | 2.49 | 27.36 | 29.81 | 18.65 | 5.28 | 169.04 | 5.58 | 12.69 | 5.23 | 426.0 | 13.21 | 3.77 |
| $T_4$ | 207.4 | 2.20 | 24.68 | 23.80 | 16.48 | 5.25 | 166.44 | 4.84 | 10.97 | 4.51 | 366.6 | 12.01 | 3.57 |
| $T_5$ | 208.4 | 2.24 | 25.03 | 24.55 | 16.93 | 5.20 | 164.04 | 4.95 | 11.04 | 4.39 | 369.4 | 12.16 | 3.62 |
| $T_6$ | 173.0 | 1.38 | 19.30 | 15.75 | 13.78 | 5.10 | 145.84 | 3.55 | 6.51 | 3.16 | 279.3 | 10.04 | 3.30 |
| $T_7$ | 194.1 | 1.78 | 21.77 | 18.75 | 15.10 | 5.30 | 153.05 | 4.60 | 8.61 | 3.67 | 313.2 | 11.21 | 3.40 |
| $T_8$ | 210.5 | 2.26 | 25.16 | 25.39 | 16.41 | 5.32 | 170.79 | 4.90 | 10.06 | 4.45 | 361.4 | 11.98 | 3.68 |
| $T_9$ | 222.4 | 2.58 | 28.46 | 29.55 | 19.74 | 5.36 | 173.52 | 5.78 | 12.97 | 5.55 | 485.1 | 13.71 | 3.92 |
| $T_{10}$ | 215.9 | 2.33 | 25.66 | 26.19 | 17.23 | 5.42 | 172.84 | 4.69 | 10.20 | 4.17 | 339.9 | 11.97 | 3.82 |
| $T_{11}$ | 230.5 | 2.70 | 28.47 | 31.54 | 21.49 | 5.50 | 178.89 | 6.26 | 13.34 | 5.59 | 514.1 | 13.92 | 4.30 |
| $T_{12}$ | 225.7 | 2.65 | 28.08 | 30.52 | 20.27 | 5.30 | 168.39 | 5.17 | 11.87 | 4.97 | 404.4 | 13.03 | 3.94 |
| SEd± | 2.79 | 0.111 | 0.251 | 0.664 | 0.744 | 0.16 | 4.03 | 0.10 | 0.745 | 0.21 | 21.1 | 0.31 | 0.096 |
| $LSD_{0.05}$ | 8.789 | 0.349 | 0.789 | 2.093 | 2.344 | 0.35 | 12.68 | 0.30 | 2.199 | 0.61 | 62.3 | 0.92 | 0.284 |

$T_1$: Rec NPK+S0, $T_2$: Rec NPK+S (40-0-0-13), $T_3$: Rec NPK+S (10-0-0-75), $T_4$: Rec NPK+AS, $T_5$: Rec NPK+Bentonite S, $T_6$: Rec PK+S (SSP), $T_7$: Rec PK+S (40-0-0-13), $T_8$: S30 1st crop ($T_2$ Source), $T_9$: Basal S 50% + Top S 50% − $T_2$ source, $T_{10}$: S30 1st crop ($T_3$ Source), $T_{11}$: Basal S 50% + Top S 50% − $T_3$ sources and $T_{12}$: $S_{60}$ 1st crop ($T_3$ Source).

*3.2. Economic Productivity*

Over NPK alone, there was a maximum increase (42 and 40%) in the pearl millet and mustard seed yield (2.60 and 2.53 t/ha, respectively) with the split application of S (50% as the base and remaining as topdressing) through SEU (10-0-0-75) along with NPK, while the pearl millet grain and mustard seed yield ranged from 1.52–2.60 and 1.71–2.53 t/ha, respectively (Table 2). The increases in seed yield of pearl millet and mustard were from −3.28 to 42.08%, and 5.52–39.78%, respectively, over NPK alone under different treatments (Figure 2). Mustard responded better in terms of seed yield to the application of S, and the response was significantly higher with SEU (40-0-0-13 and 10-0-0-75) than with other sources of S, including ammonium sulfate (AS), bentonite S, and SSP. Stover and biological yields were also recorded as being significantly higher under split application of SEU (10-0-0-75) ($T_{11}$) with half as the base and the remainder as topdressing, but these remained statistically on par with $T_9$ and $T_{12}$, where all of the S was applied from the $T_2$ and $T_3$ S sources, respectively. However little variation, even in stover and biological yield, was recorded in the mustard crop, where the $T_3$ source of S application resulted in higher vegetative biomass, but it remained on par with $T_{11}$. There was no effect on the harvest index of either crop (pearl millet as well as mustard) under the pearl millet–mustard system. The lowest seed yield was recorded in $T_6$ (1.52 t/ha and 1.71 t/ha for pearl millet and mustard, respectively). Correlation analyses of the growth, yield attributes, oil content, and seed and oil yield were conducted for mustard to find the interrelationship. The data revealed that plant height, primary and secondary branches, and total siliqua/plant have a positive and significant correlation with seed and oil yield but not with oil content. Oil yield was significantly positively correlated with most of the parameters under study, but oil content was observed to be positively correlated with secondary branches per plant, siliqua length, and 1000 seed weight (Table 3).

**Table 2.** Effect of S-enhanced urea and other S sources on yield and harvest index of pearl millet and mustard in pearl millet–mustard system.

| Treatment | Pearlmillet | | | | Mustard | | | |
|---|---|---|---|---|---|---|---|---|
| | Grain Yield (t/ha) | Stover Yield (t/ha) | Biological Yield (t/ha) | HI | Seed Yield (t/ha) | Stover Yield (t/ha) | Biological Yield (t/ha) | HI |
| $T_1$ | 1.83 | 6.60 | 8.43 | 21.5 | 1.81 | 6.57 | 8.60 | 24.2 |
| $T_2$ | 2.28 | 7.89 | 10.17 | 22.4 | 2.30 | 7.20 | 9.53 | 24.7 |
| $T_3$ | 2.36 | 8.15 | 10.50 | 22.4 | 2.44 | 7.77 | 10.17 | 24.4 |
| $T_4$ | 2.02 | 7.47 | 9.48 | 21.1 | 2.24 | 6.67 | 8.93 | 27.8 |
| $T_5$ | 2.04 | 7.56 | 9.60 | 21.1 | 2.11 | 7.30 | 9.57 | 24.4 |
| $T_6$ | 1.52 | 5.66 | 7.17 | 21.0 | 1.71 | 6.47 | 8.20 | 23.8 |
| $T_7$ | 1.77 | 6.38 | 8.14 | 21.5 | 2.21 | 6.70 | 8.73 | 23.9 |
| $T_8$ | 2.14 | 7.54 | 9.66 | 22.0 | 2.18 | 7.13 | 9.33 | 24.5 |
| $T_9$ | 2.54 | 8.53 | 11.07 | 22.9 | 2.52 | 7.77 | 10.27 | 24.8 |
| $T_{10}$ | 2.16 | 7.84 | 10.00 | 21.5 | 2.27 | 6.63 | 8.93 | 27.9 |
| $T_{11}$ | 2.60 | 8.59 | 11.19 | 23.2 | 2.53 | 7.47 | 10.10 | 28.0 |
| $T_{12}$ | 2.52 | 8.53 | 11.05 | 22.7 | 2.32 | 7.73 | 10.13 | 24.3 |
| SEd± | 0.052 | 0.171 | 0.20 | 0.642 | 0.065 | 0.325 | 0.34 | 2.053 |
| $LSD_{0.05}$ | 0.163 | 0.537 | 0.63 | NS | 0.193 | 0.959 | 1.003 | NS |

$T_1$: Rec NPK+S0, $T_2$: Rec NPK+S (40-0-0-13), $T_3$: Rec NPK+S (10-0-0-75), $T_4$: Rec NPK+AS, $T_5$: Rec NPK+Bentonite S, $T_6$: Rec PK+S (SSP), $T_7$: Rec PK+S (40-0-0-13), $T_8$: S30 1st crop ($T_2$ Source), $T_9$: Basal S 50% + Top S 50% − $T_2$ source, $T_{10}$: S30 1st crop ($T_3$ Source), $T_{11}$: Basal S 50% + Top S 50% − $T_3$ sources and $T_{12}$: $S_{60}$ 1st crop ($T_3$ Source), HI: Harvest Index.

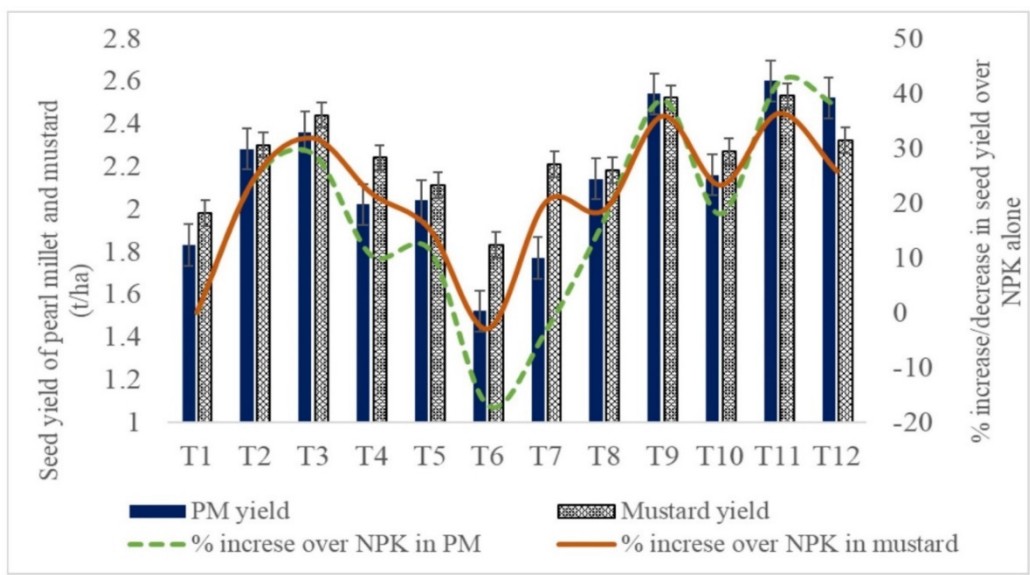

**Figure 2.** Mustard crop yield as influenced by different S-enhanced ureas and other S sources. $T_1$: Rec NPK+S0, $T_2$: Rec NPK+S (40-0-0-13), $T_3$: Rec NPK+S (10-0-0-75), $T_4$: Rec NPK+AS, $T_5$: Rec NPK+Bentonite S, $T_6$: Rec PK+S (SSP), $T_7$: Rec PK+S (40-0-0-13), $T_8$: S30 1st crop ($T_2$ Source), $T_9$: Basal S 50% + Top S 50% − $T_2$ source, $T_{10}$: S30 1st crop ($T_3$ Source), $T_{11}$: Basal S 50% + Top S 50% − $T_3$ sources and $T_{12}$: $S_{60}$ 1st crop ($T_3$ Source).

**Table 3.** Pearson correlation matrix for Indian mustard.

| Parameters | PH | Pb | Sb | Siliqua Length | Total Siliqua | Siliqua Length | 1000 Seed Weight | Seed Yield | Oil Content | Oil Yield |
|---|---|---|---|---|---|---|---|---|---|---|
| PH | 1.00 | | | | | | | | | |
| Pb | 0.84 ** | 1.0 | | | | | | | | |
| Sb | 0.85 ** | 0.95 ** | 1.0 | | | | | | | |
| Siliqua length | 0.82 ** | 0.95 ** | 0.98 ** | 1.000 | | | | | | |
| Total siliqua | 0.79 ** | 0.96 ** | 0.94 ** | 0.966 ** | 1.0 | | | | | |
| Siliqua length | 0.86 ** | 0.98 ** | 0.98 ** | 0.982 ** | 0.97 ** | 1.0 | | | | |
| 1000 seed weight | 0.84 ** | 0.84 ** | 0.85 ** | 0.846 ** | 0.88 ** | 0.88 ** | 1.0 | | | |
| Seed yield | 0.77 ** | 0.88 ** | 0.91 ** | 0.887 ** | 0.85 ** | 0.90 ** | 0.84 ** | 1.0 | | |
| Oil content | 0.38 NS | 0.52 NS | 0.59 * | 0.605 * | 0.57 NS | 0.60 * | 0.70 * | 0.75 ** | 1.0 | |
| Oil yield | 0.86 ** | 0.93 ** | 0.97 ** | 0.951 ** | 0.92 ** | 0.97 ** | 0.90 ** | 0.95 ** | 0.70 * | 1.000 |

PH: Plant height; Pb: Primary branches; Sb: Secondary braches; NS: Non-significant; * Significant at 5% probability level; ** Significant at 1% probability level.

### 3.3. Oil Content, Oil Yield, and Fatty Acid Composition in Mustard

Oil content and oil yield in mustard seed were influenced significantly by different S sources and their method of application. The maximum oil content was observed in $T_{11}$ (basal S 50% + top S 50% − $T_3$ sources and NPK), while it remained statistically on par with the remaining sources of S. However, overall, S application irrespective of the source and method resulted in significantly higher oil content compared to the oil content in the mustard crop under NPK alone and without S application. However, the trend in oil yield (kg/ha) was variable (Table 4), directly coinciding with the seed yield of the mustard crop. SEU (40-0-013 and 10-0-0-75) resulted in significantly higher oil yield over NPK alone. The maximum oil yield was recorded under the split application of S ($T_9$ and $T_{11}$) from SEU (40-0-0-13 and 10-0-0-75) sources along with the recommended NPK. The composition of fatty acids in mustard oil (Table 5 and Figure 3) is of paramount importance, and there was a significant effect of S sources on the mono- and polyunsaturated (MUFA and PUFA) and saturated fatty acid (SFA) compositions of mustard oil under different treatments. The SFA composition ranged from 5.2–10.9%, while the MUFA and PUFA compositions ranged from 60.7 to 74.0% and 15.1 to 31.2%, respectively. Imposition of the recommended NPK+AS ($T_4$) and basal S 50% + top S 50% − $T_3$ ($T_{11}$) sources resulted in a lower SFA composition (5.2

and 5.9%, respectively), but the basal S 50% + top S 50% − $T_2$ source and the recommended NPK+S (10-0-0-75) increased the SFA composition, with maximum values of 10.9 and 10.6% respectively. The MUFA composition remains very high in mustard oil, and the $T_9$: basal S 50% + top S 50% − $T_2$ source resulted in the maximum MUFA (74.0%) among the seeds, whereas the maximum PUFA was found in the seed grown under $T_8$: $S_{30}$ 1st crop ($T_2$). Therefore, the SEU and other S sources influenced the fatty acid composition differently (Figure 4).

**Table 4.** Fatty acid compositions of the mustard seeds as influenced by different S sources.

| Treatment | Oil Content (%) | Oil Yield (kg/ha) | Fatty Acid Composition (%) | | |
|---|---|---|---|---|---|
| | | | SFA | MUFA | PUFA |
| $T_1$: Rec NPK+$S_0$ | 38.2 | 773.1 | 8.9 | 67.1 | 23.9 |
| $T_2$: Rec NPK+S (40-0-0-13) | 39.1 | 912.0 | 8.1 | 60.7 | 31.2 |
| $T_3$: Rec NPK+S (10-0-0-75) | 39.9 | 954.9 | 10.6 | 66.9 | 22.5 |
| $T_4$: Rec NPK+AS | 39.2 | 890.3 | 5.2 | 69.5 | 22.5 |
| $T_5$: Rec NPK+Bentonite S | 38.7 | 855.5 | 8.3 | 67.9 | 20.9 |
| $T_6$: Rec PK+S (SSP) | 39.0 | 679.3 | 9.0 | 61.4 | 29.6 |
| $T_7$: Rec PK+S (40-0-0-13) | 39.4 | 808.5 | 9.2 | 70.5 | 20.3 |
| $T_8$: $S_{30}$ 1st crop ($T_2$ source) | 39.2 | 857.4 | 7.9 | 61.2 | 30.9 |
| $T_9$: Basal S 50% + top S 50% − $T_2$ source | 39.7 | 1018.0 | 10.9 | 74.0 | 15.1 |
| $T_{10}$: $S_{30}$ 1st crop ($T_3$ Source) | 39.3 | 888.1 | 6.2 | 67.2 | 26.6 |
| $T_{11}$: Basal S 50% + top S 50% − $T_3$ sources | 39.8 | 1014.2 | 5.9 | 67.6 | 26.5 |
| $T_{12}$: $S_{60}$ 1st crop ($T_3$ Source) | 40.3 | 964.7 | 7.6 | 67.9 | 24.5 |
| SEd± | 0.31 | 28.04 | 0.35 | 0.28 | 1.06 |
| $LSD_{0.05}$ | 0.92 | 82.77 | 0.75 | 0.62 | 2.12 |

Rec: Recommend dose; SEd±: Standard Error of Mean; LSD: Least significant difference; SFA: saturated fatty acids; MUFA: monounsaturated fatty acids; PUFA: polyunsaturated fatty acids.

**Table 5.** Effect of S-enhanced urea and other S sources on nutrient uptake by pearl millet and mustard.

| Treatment | Pearlmillet | | Mustard | | Pearlmillet-Mustard System | |
|---|---|---|---|---|---|---|
| | N Uptake by Grain (kg/ha) | S Uptake by Grain (kg/ha) | N Uptake by Seed (kg/ha) | S Uptake by Seed (kg/ha) | Total N Uptake kg/ha (Seed) | Total S Uptake kg/ha (Seed) |
| $T_1$ | 29.8 | 4.1 | 65.1 | 23.6 | 94.9 | 27.7 |
| $T_2$ | 39.1 | 7.1 | 78.2 | 33.5 | 117.3 | 40.6 |
| $T_3$ | 40.6 | 7.7 | 89.3 | 36.2 | 129.9 | 43.9 |
| $T_4$ | 33.7 | 5.7 | 75.0 | 32.5 | 108.6 | 38.1 |
| $T_5$ | 33.9 | 5.8 | 72.1 | 31.9 | 105.9 | 37.7 |
| $T_6$ | 22.8 | 4.1 | 49.7 | 21.3 | 72.5 | 25.4 |
| $T_7$ | 27.8 | 4.8 | 71.0 | 28.3 | 98.8 | 33.1 |
| $T_8$ | 36.0 | 5.8 | 70.2 | 28.1 | 106.1 | 33.8 |
| $T_9$ | 44.6 | 8.4 | 94.1 | 38.7 | 138.6 | 47.0 |
| $T_{10}$ | 36.5 | 6.2 | 79.0 | 31.2 | 115.4 | 37.4 |
| $T_{11}$ | 45.8 | 9.1 | 100.5 | 40.3 | 146.2 | 49.4 |
| $T_{12}$ | 43.5 | 8.8 | 79.1 | 34.4 | 122.6 | 43.2 |
| SEd± | 0.704 | 0.476 | 3.428 | 1.377 | 3.716 | 1.61 |
| $LSD_{0.05}$ | 2.216 | 1.498 | 10.797 | 4.339 | 11.705 | 5.07 |

$T_1$: Rec NPK+S0, $T_2$: Rec NPK+S (40-0-0-13), $T_3$: Rec NPK+S (10-0-0-75), $T_4$: Rec NPK+AS, $T_5$: Rec NPK+Bentonite S, $T_6$: Rec PK+S (SSP), $T_7$: Rec PK+S (40-0-0-13), $T_8$: S30 1st crop ($T_2$ Source), $T_9$: Basal S 50% + Top S 50% − $T_2$ source, $T_{10}$: S30 1st crop ($T_3$ Source), $T_{11}$: Basal S 50% + Top S 50% − $T_3$ sources and $T_{12}$: $S_{60}$ 1st crop ($T_3$ Source), N: Nitrogen; S: Sulfur.

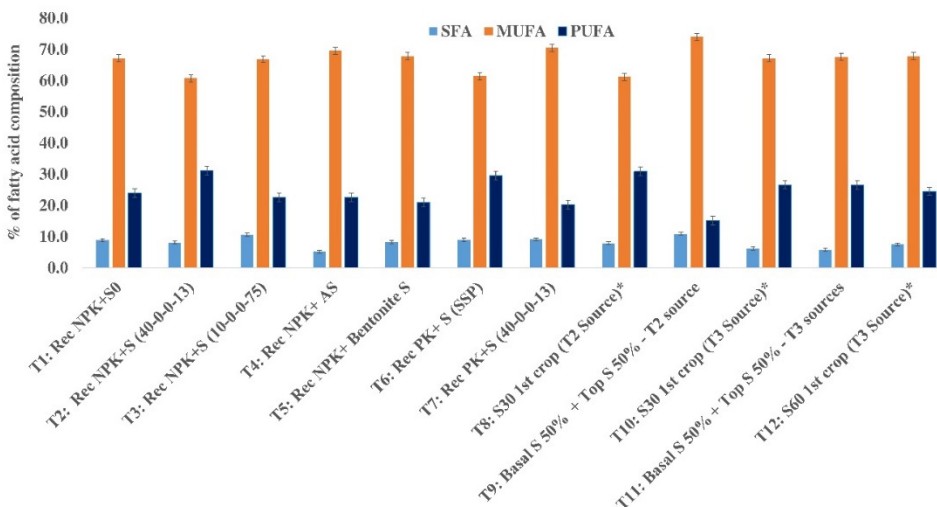

**Figure 3.** Fatty acid composition (saturated fatty acids: SFA, monounsaturated fatty acids: MUFA, polyunsaturated fatty acids: PUFA) of mustard oil as influenced under different S-enhanced urea and other S sources in the pearl millet–mustard system. * Residual effect.

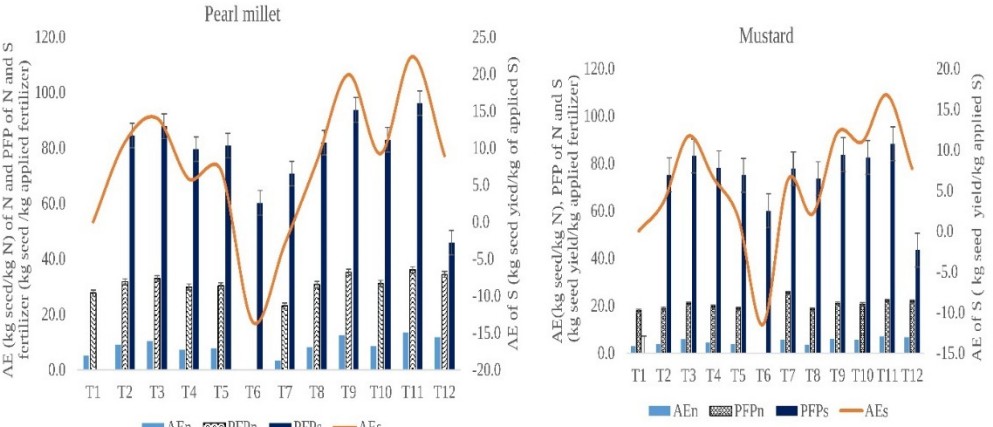

**Figure 4.** AEn, AEs, PFPn, and PFPs of pearl millet and mustard crops as influenced under different S enhanced urea and other S sources in pearl millet–mustard system. $T_1$: Rec NPK+S0, $T_2$: Rec NPK+S (40-0-0-13), $T_3$: Rec NPK+S (10-0-0-75), $T_4$: Rec NPK+AS, $T_5$: Rec NPK+ Bentonite S, $T_6$: Rec PK+S (SSP), $T_7$: Rec PK+S (40-0-0-13), $T_8$: S30 1st crop ($T_2$ Source), $T_9$: Basal S 50% + Top S 50% − $T_2$ source, $T_{10}$: S30 1st crop ($T_3$ Source), $T_{11}$: Basal S 50% + Top S 50% − $T_3$ sources and $T_{12}$: $S_{60}$ 1st crop ($T_3$ Source). AEn: Agronomic efficiency of nitrogen; PFPn: Partial factor productivity of nitrogen; AEs: Agronomic efficiency of nitrogen; PFPs: Partial factor productivity of nitrogen.

### 3.4. Nutrient Usage and Soil Properties

Among the S sources, the imposition of $T_{11}$ (basal S 50% + top S 50% − $T_3$ sources) resulted in significantly higher N and S uptake by grain (146.2 and 49.4 kg/ha), which remained statistically on par (138.6 and 47.0 kg/ha, respectively) with $T_9$ (basal S 50% + top S 50% − $T_2$ source) (Table 6). With regard to total N uptake by the system, while comparing the other treatments, $T_2$ and $T_{12}$ were found to be equally effective and remained superior to treatments that consisted of NPK ($T_1$) and PKS only ($T_6$). The N uptake by seeds of the system was found to be lower with the application of $T_6$. The application of $T_3$ failed to produce a significant difference from $T_2$ in the case of grain S uptake and produced higher seed S uptake as compared to $T_6$, which exhibited the lowest seed S uptake by a significant amount. However individually, the pearl millet crop removed significantly more N (45.78 kg/ha) from $T_{11}$ (basal S 50% + top S 50% − $T_3$ source), which showed a similar result as in the case of S uptake (9.12 kg/ha) by pearl millet. Similar results were

also obtained in the mustard crop, with significantly higher accumulation of N in seeds under $T_{11}$ as compared to $T_6$; also similar was the trend in S uptake through seeds in the mustard crop. In mustard, $T_2$ and $T_3$ resulted in almost similar amounts of S uptake as $T_{12}$. In the pearl millet–mustard system, $T_{11}$ and $T_9$ resulted in significantly higher N and S uptake by seeds.

**Table 6.** Effect of S-enhanced urea and other S sources on soil chemical properties at flowering and harvest.

| Treatment | EC (dS/m) | | pH | | OC % | | N (kg/ha) | | S (mg/kg) | |
|---|---|---|---|---|---|---|---|---|---|---|
| | Initial | Final | Initial | Final | Initial | Final | Flowering | Harvest | Flowering | Harvest |
| $T_1$ | 0.29 | 0.28 | 7.94 | 7.77 | 0.46 | 0.48 | 179.85 | 165.5 | 13.5 | 12.1 |
| $T_2$ | 0.34 | 0.30 | 7.86 | 7.72 | 0.51 | 0.46 | 207.62 | 185.2 | 19.98 | 20.0 |
| $T_3$ | 0.35 | 0.30 | 7.85 | 7.72 | 0.51 | 0.47 | 213.04 | 183.7 | 20.04 | 23.6 |
| $T_4$ | 0.32 | 0.30 | 7.92 | 7.87 | 0.48 | 0.48 | 194.07 | 172.1 | 18.74 | 21.6 |
| $T_5$ | 0.34 | 0.30 | 7.88 | 7.80 | 0.49 | 0.49 | 196.92 | 170.3 | 19.03 | 21.8 |
| $T_6$ | 0.28 | 0.29 | 8.05 | 7.87 | 0.44 | 0.48 | 154.71 | 124.5 | 15.73 | 21.0 |
| $T_7$ | 0.29 | 0.31 | 7.98 | 7.61 | 0.45 | 0.50 | 171.35 | 151.5 | 15.87 | 22.1 |
| $T_8$ | 0.30 | 0.31 | 7.93 | 7.78 | 0.49 | 0.48 | 200.65 | 178.5 | 14.98 | 22.6 |
| $T_9$ | 0.35 | 0.32 | 7.81 | 7.69 | 0.52 | 0.51 | 214.13 | 197.5 | 20.35 | 24.8 |
| $T_{10}$ | 0.31 | 0.32 | 7.93 | 7.67 | 0.50 | 0.51 | 204.19 | 175.2 | 15.12 | 20.7 |
| $T_{11}$ | 0.36 | 0.34 | 7.78 | 7.66 | 0.53 | 0.53 | 222.92 | 201.0 | 21.02 | 23.6 |
| $T_{12}$ | 0.36 | 0.33 | 7.78 | 7.53 | 0.52 | 0.48 | 218.01 | 190.4 | 21.65 | 24.9 |
| SEd± | 0.03 | 0.01 | 0.16 | 0.05 | 0.03 | 0.006 | 7.98 | 5.1 | 0.81 | 0.7 |
| $LSD_{0.05}$ | NS | 0.02 | NS | 0.16 | NS | 0.019 | 16.55 | 15.1 | 1.69 | 2.1 |

$T_1$: Rec NPK+S0, $T_2$: Rec NPK+S (40-0-0-13), $T_3$: Rec NPK+S (10-0-0-75), $T_4$: Rec NPK+AS, $T_5$: Rec NPK+Bentonite S, $T_6$: Rec PK+S (SSP), $T_7$: Rec PK+S (40-0-0-13), $T_8$: S30 1st crop ($T_2$ Source), $T_9$: Basal S 50% + Top S 50% − $T_2$ source, $T_{10}$: S30 1st crop ($T_3$ Source), $T_{11}$: Basal S 50% + Top S 50% − $T_3$ sources and $T_{12}$: $S_{60}$ 1st crop ($T_3$ Source).

Data about soil properties at the initial stage (before the sowing of pearl millet) and after the harvest of the mustard crop revealed significant responses to different S sources (Table 5). The results revealed that there was significant variation in available N and S at flowering as well as at harvest, while soil pH, EC, and OC at harvest were noted as being non-significant due to different treatments. Available N was found to be significantly higher under $T_{11}$ and remained statistically similar to $T_{12}$, $T_9$, $T_3$, and $T_2$ at flowering, but at harvest, it was found to be on par with $T_{12}$ and $T_9$ only. The soil-available S was improved with different fertilization of S compared to without S application. The maximum available S was recorded with $T_{12}$, which significantly differed from other treatments except for $T_{11}$, $T_9$, $T_3$, and $T_2$ at flowering. At harvest, treatment $T_{12}$ recorded a significantly higher value of available S, which was found to be on par with treatments $T_{11}$, $T_9$, and $T_3$. However, the exclusion of S resulted in lower values of soil-available S (13.5 and 12.1 mg/kg) at flowering and harvest, respectively.

### 3.5. Agronomic Use Efficiency and Partial Factor Productivity of N and S

The agronomic efficiency (AEn) of N increased substantially under supplementation of S through SEU (40-0-0-13 and 10-0-0-75). Even SEU resulted in a higher AEn of N compared to other S sources, i.e., bentonite S, SSP, and ammonium sulfate. The splitting of sulfur doses through SEU of 40-0-0-13 and 10-0-0-75 enhanced the AEn of pearl millet to the maximum level (12.5 and 13.5 kg seed/kg of nutrient applied, respectively) over NPK alone (27.7 kg seed yield/kg applied N fertilizer). Compared to the PFPn, the PFPs was higher for almost all the nutrient management options ($T_2$ to $T_{12}$). The PFPs ranged from −96.1 kg seed yield/kg applied S fertilizer. The splitting of S sources through SEU of 40-0-0-13 and 10-0-0-75 resulted in the highest PFPs (93.7 and 96.1 kg seed yield/kg applied S fertilizer, respectively) in pearl millet. With 10.8 and 14.0 kg seed yield/kg S applied, the splitting of SEU sources in $T_9$ and $T_{11}$ also resulted in enhanced AEn of S (19.9 and 22.3 kg seed yield/kg of applied S, respectively). In the case of mustard, the partial factor

productivity of SEU also remained higher—at 31.7 and 23.9 with 40-0-0-13 and 10-0-0-75, respectively—over NPK alone (27.7 kg seed yield/kg applied N fertilizer). Compared to the PFPn, the PFPs values were higher for almost all the nutrient management options ($T_2$ to $T_{12}$). The PFPs ranged from 60.2–96.1 kg seed yield/kg applied S fertilizer. The splitting of S sources through SEU of 40-0-0-13 and 10-0-0-75 resulted in the highest PFPs values (93.7 and 96.1 kg seed yield per kg applied S fertilizer, respectively).

## 4. Discussion

The effect of sulfur (S) on the seed yield and quality of oilseeds and cereals has been explained previously [5,11,21–23]. However, the optimum level, sources, and timing of sulfur usage in pearl millet–mustard systems under rainfed conditions remain largely unclear. This experiment intended to evaluate the response of pearl millet–mustard to different S sources and levels and their effect on seed yield and quality (oil contents and oil quality), S uptake, and soil properties. The results of the field experiments revealed that in both crops (pearl millet and mustard), the growth- and yield-related parameters were improved with sulfur application (Table 1). This increase might be due to the availability and utilization of essential nutrients (N, P, and K), as these nutrients increased with S application [24] due to its synergistic effects. The findings showed that a higher rate of S application, i.e., 40 kg/ha, improved seed yield due to the considerable increase in yield attributing parameters (Table 2). S deficiency affects the growth, development, disease resistance, and performance of plants and has a great impact on the nutritional quality of crops [25]. The primary and dominant S source is inorganic sulfate ($SO_4^{2-}$) for plant growth. The chloroplasts of young leaves are the prominent organelle where assimilation of $SO_4^{2-}$ by cysteine occurs; however, the synthesis of methionine and cysteine can also happen in seeds and roots [26]. Adequate S management in a pearl millet–mustard system may be the reason for the better response to applied S sources over NPK alone. Singh et al. [5] highlighted widespread nutrient deficiencies, especially of S, as the major soil health problem for sustaining crop productivity in many parts of the northwestern Indo-Gangetic Plain (IGP) and the Western Himalayan Region (WHR). Because S is an essential constituent of enzymes involved in N metabolism, i.e., nitrate reductase and nitrite reductase [15,27,28], its deficiency could lead to a decrease in N assimilation. Some reports have also shown that nitrogen with S synergistically enhances crop growth, productivity, and the agronomic use efficiency of N and S [5,11,29,30]. A strong interaction of S and N for seed yield was found in rapeseed and mustard [11,31,32], sunflower [33], linseed [34] groundnut [35], and soybean [36]. Aulakh et al. reported that the maximum grain yield in mustard was obtained with 30 kg S/ha supplied as gypsum along with 120 kg N/ha as urea. Hence, a combination of N and S in SEU lead to a better response [37].

Oilseed rape has high S requirements, especially from the budding stage to the siliqua-forming stage, which may be the reason for the maximum crop growth and productivity occurring under the split application of SEU [38–40]. The S availability during this period ensures the proper growth and development of oilseed rape. Janzen and Bettany also claim that sulfur fertilizers fundamentally affect crop yield [41]. The authors observed particularly favorable effects of fertilizer applied at the budding stage. If S deficiencies are observed during the growing season, application of sulfate S during the period from the start of flowering may be beneficial, although yields will generally be lower than if S is available during flower initiation [42]. In SEU, the micronization of elemental S into small particles (on average smaller than 75 μm, or 75/1000 mm) to promote oxidation to sulfate within the growing season, offering S availability across the crop season, may be the reason for the better response in terms of growth and yield of the mustard and pearl millet crops over other sources of S [43]. The combination of slow-released elemental S and immediately available sulfate S in SEU helps to provide season-long S access to crops. Synchronous release of S from SEU over the crop growth period resulted in high oil content and subsequently increased oil yield. Enhanced seed yield under adequate S nutrition is

mainly responsible for higher oil yield [44]. The results of the current study were congruent with past works conducted by various researchers [45,46].

S and boron application markedly reduced the saponification of saturated fatty acids (palmitic and stearic) and increased the unsaturated fatty acids (linoleic and oleic) and iodine value [33]. The results of a past study also highlighted that manipulating the S supply might be one means of altering the fatty acids profile among the brassica genotypes [47]. S facilitates the conversion of carbohydrates via oil and fatty acid synthesis [48]. Furthermore, the activities of thiokinase (which plays a significant role in oil synthesis) are also mediated by S application [48]. Even N fertilizer, when applied in combination with S, also influenced the fatty acid composition in oilseed brassica [49]. Kumar et al. also explained the nutritional factors that govern the fatty acid composition of rapeseed mustard oil [50]. A progressive increase in S doses (S 30 and S 45) correspondingly increases the proportion of saturated fatty acids in vegetable oils [51]. The composition of the analyzed monounsaturated fatty acids did not change significantly after applying S fertilizer, although for all the doses of S, there was a relevant decrease in the proportion of polyunsaturated acids [52,53]. A decrease in the content of oleic and palmitic acids in the seeds of rape after fertilizing with either the sulfate or elemental form of S was reported by Szulc et al. [54]. Application of S in spring rape improved the nutritional value of oil by significantly increasing the proportion of essential unsaturated fatty acids (C18:2 and C18:3) [55,56]. Jan et al. also noted that sulfur fertilization significantly increased the proportion of erucic acid in rape seeds [53].

S application enhances nitrogen use efficiency under N-stressed conditions. Nitrogen use efficiency, i.e., kg grain per unit N rate, was 50% higher when the crop received S fertilizer over no S application [13]. Previous studies also recorded the lowest values of fat content in the treatment where sulfur had not been applied [57,58]. Furthermore, Malhi et al. further reported a maximized oil yield of the oilseed brassica at 30 kg S/ha [58].

S and N have a synergetic effect; hence S addition improves N use efficiency. However, a corresponding increase in S doses failed to affect the N uptake significantly. This interaction was also observed when analyzing biomass and grain yield [59] and in previously reported studies of S fertilization in grasses [60]. A synergistic as well as antagonistic relationship between N and S use efficiency at optimal and excessive levels, respectively has been reported by Fismes et al. [61] and Kumar et al. [62]. Nutrient deficiency at the vegetative and reproductive stages can have a detrimental effect on the overall growth potential of the crops. This was the reason for the decline in crop yield and $AE_N$ and $AE_S$.

The increase in biomass observed in response to S addition by Salvagiotti and Miralles [59] can be explained in terms of a greater accumulation of N in vegetative tissues, i.e., stored as rubisco in leaves. As no change in grain N concentration was observed, the accumulation of N in the seed may have been proportional to biomass production. Then, the larger N accumulation in the grains due to S addition was explained by increases in seed mass, rather than changes in grain N concentration. The experimental results proved that sulfur application at a rate of 80–100 kg S/ha increased the concentrations of S and N in the seeds and stover of oilseed rape [40,63]. The simultaneous increase in the dry matter of rape significantly increased the total accumulation of the elements tested relative to the control plants. A wide range (from 0.02 to 822 mg/kg) of different states has been reported by Shukla et al. [21]. These multi-nutrient deficiencies could be alleviated by the production, distribution, and application of S and micronutrients containing customized fertilizers prepared based on prevailing nutrient deficiencies in different agroecologies. This will help to a greater extent in maintaining soil health, achieving sustainable crop production, and attaining better quality of crops [21,64,65].

The variation in the soil-available S is mainly ascribed to differences in the acreage of S-loving crops and less or no addition of S-containing fertilizers. The deficiency of available S could be efficiently alleviated by adopting site-specific S manipulation strategies in various soil–crop contexts. Adequate S fertilization is also suggested by many researchers to overcome the S deficiency in different soil–crop contexts in various regions of

India [5,21,47,66,67]. Soil properties like available N, available S, and soil EC, pH, and OC changed with different S sources over the experimentation period. The highest availability of inorganic S was recorded in soil that received 20 mg S and 10 gm farmyard manure (FYM) per kg soil. Many soils have an accumulation of adsorbed sulfate in the subsoil because conditions (low pH and high clay) are favorable for adsorption [31,67], and the higher pH under experimental soil may lead to lesser adsorption and subsequently lesser availability. Kulczycki and Muhammad et al. also highlighted scanty variation in soil chemical properties due to S sources [31,51]. However, other studies report a greater concentration and accumulation of sulfate after S fertilization with sulfates [37], as well as with elemental S [31]. Fertilizing mustard with S also had a positive impact on the level of N in the seeds [51], the leaf area index, the rate of photosynthesis, and biomass production in mustard.

## 5. Conclusions

The study spells out the worthiness of the new unique sulfonated urea fertilizer sources of S and N in enhancing growth, yield attributing traits, and seed yield in the pearl millet–mustard cropping system. The agronomic use efficiency of N and S also improved significantly under the usage of SEU sources (10-0-0-75 and 40-0-0-13). The oil content and oil yield, along with the favorable impact on fatty acid composition, were also observed. The results of the current study suggest that the split application of SEU 15.0 kg/ha (50%) as the base and 15.0 kg/ha as the topdressing (50%) is imperative for profitable and soil-supportive cultivation of the pearl millet–mustard cropping system in a semi-arid agroecology.

**Author Contributions:** Conceptualization, S.S.R.; Data curation, K.S.; Formal analysis, P.K.U. and K.C.S.; Investigation, S.S.R., S.B. and M.H.; Methodology, S.B., K.S. and M.H.; Project administration, V.K.S.; Resources, S.S.R., V.K.S., R.K.S., M.H., K.C.S. and R.S.; Software, R.K.S.; Supervision, V.K.S.; Writing—original draft, S.B. and R.J.; Writing—review & editing, K.S., R.K.S., P.K.U., R.J. and R.S. All authors have read and agreed to the published version of the manuscript.

**Funding:** This research was funded by the Shell India Markets Pvt. Ltd.

**Institutional Review Board Statement:** This research was conducted as per the relevant guidelines of the Institute.

**Informed Consent Statement:** Not applicable.

**Data Availability Statement:** The data that support the findings of this study were statistically analyzed and are presented in the manuscript.

**Conflicts of Interest:** The authors declare no conflict of interest.

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
