# Peer review of "Sulfur Sources Mediated the Growth, Productivity, and Nutrient Acquisition Ability of Pearlmillet–Mustard Cropping Systems"

_sustainability, doi:10.3390/su142214857_

Round 1

Reviewer 1 Report

The author has explored the impact of soil on productivity. On the whole, this article is of great value, but I think this article has several important documents that need to be cited. After careful revision by the author, this article can be considered for acceptance.

  • DOI: 
  • 10.3389/fpls.2022.996313
  • DOI: 
  • 10.3390/plants11162171
  • DOI: 
  • 10.3389/fpls.2022.990441

Author Response

S. No.

Specific Comments

Compliance

1

The author has explored the impact of soil on productivity. On the whole, this article is of great value, but I think this article has several important documents that need to be cited. After careful revision by the author, this article can be considered for acceptance.

Respected reviewer thanks you very much for appreciating the works. We have incorporated all your suggestions. Your suggestions helped us a lot to improve the manuscript.  Suggested references (DOI:  10.3389/fpls.2022.996313 DOI:  10.3390/plants11162171 DOI:  10.3389/fpls.2022.990441) added in the manuscript

-----

Reviewer 2 Report

The manuscript must be improved. Other suggestions can be found in the text.

Author Response

FS. No.

Specific Comments

Compliance

1

The manuscript must be improved. Other suggestions can be found in the text.

Respected reviewer thanks you very much for appreciating the works. We have incorporated all your suggestions. Your suggestions helped us a lot to improve the manuscript.

2.

How was the statistical analysis was done

Statistical analysis procedure added

3.

Add geographical coordinates and altitudes

Necessary corrections made in revised manuscript.

4.

What does the acronym means

All the acronyms spelled out where ever required.

5.

Add extractors of soil chemical attributes in addition to the sampling depth

Suggestion incorporated in revised manuscript

6.

Where did you base the dose S applied?

Criteria for choosing treatments

Which variables are evaluated

Cultural tracts used for crops

Based on the soil test value performed before the conducting of experiment. To find out the best source these treatments were selected and recommend cultural practices for pearlmillet and mustard were used to raise the crops.

7.

What does the acronym means

All the acronyms spelled out where ever required.

8.

What does the acronym means

All the acronyms spelled out where ever required.

9

What does the acronym means

All the acronyms spelled out where ever required.

10

What was the purpose for performing the linear correlations

The linear correlation coefficient was used to determine the magnitude and direction of a linear link between the variables.

11

The analyze of S and N were not related in materials and methods

Added in m& m section

12

Analyze not related in materials and methods

Added in m& m section

Reviewer 3 Report

Correções descritas no arquivo anexado

Author Response

v=

FS. No.

Specific Comments

Compliance

1

correções descritas no arquivo anexado

Respected reviewer thanks you very much for appreciating the works. We have incorporated all your suggestions. Your suggestions helped us a lot to improve the manuscript.

2.

Better describe and report on water balance

Respected reviewer thanks you very much for your valuable suggestions, but we have not measured water balance in this study. Now the experiment is over and at this juncture, it is not possible to add it. But your point is very genuine we will take care of it in our future and current research planning.

3.

Please correct

Necessary corrections were made in the revised manuscript.

4.

Please check value and score

Corrected

Round 2

Reviewer 1 Report

I am very happy to accept this article now.